# How to Tackle Discordance in Adjuvant Chemotherapy Recommendations by Using Oncotype DX Results, in Early-Stage Breast Cancer

**DOI:** 10.3390/cancers16172928

**Published:** 2024-08-23

**Authors:** Katalin Boér, Ambrus Kaposi, Judit Kocsis, Zsolt Horváth, Balázs Madaras, Ákos Sávolt, Gyorgy Benjamin Klément, Gábor Rubovszky

**Affiliations:** 1Department of Medical Oncology, Szent Margit Hospital, 1032 Budapest, Hungary; dr.boer.katalin@sztmargit.hu; 2Department of Programming Languages and Compilers, Faculty of Informatics, Eötvös Loránd University (ELTE), 1117 Budapest, Hungary; akaposi@inf.elte.hu; 3Department of Oncoradiology, Bács-Kiskun County Hospital, 6000 Kecskemét, Hungary; kocsisj@kmk.hu (J.K.); horvathzso@kmk.hu (Z.H.); 4Department of Thoracic and Abdominal Tumors and Clinical Pharmacology, National Institute of Oncology, 1122 Budapest, Hungary; madaras.balazs@oncol.hu (B.M.); klement.gyorgy@oncol.hu (G.B.K.); 5Department of Breast and Sarcoma Surgery, National Institute of Oncology, 1122 Budapest, Hungary; 6National Tumor Biology Laboratory, 1122 Budapest, Hungary; 7Department of Oncology, Semmelweis University, 1122 Budapest, Hungary

**Keywords:** breast cancer, adjuvant chemotherapy, genetic predictive testing, decision-making

## Abstract

**Simple Summary:**

Postoperative adjuvant chemotherapy generally improves survival in patients with breast cancer. However, adjuvant chemotherapy does not benefit all patients. There are considerations and guidelines that guide us as to whether or not chemotherapy is recommended to a particular patient. The decision is based on clinicopathologic features and may be aided by multigene assays. The Oncotype DX test is used worldwide. It makes the recommendation more accurate; however, there are possibilities to refine the process to make a more accurate decision. We investigated how we could move forward in recommending adjuvant chemotherapy.

**Abstract:**

Background: The use of the Oncotype DX test reduces the rate of adjuvant chemotherapy recommendations. Few in-depth analyses have been performed on this decision-making process. Methods: We retrospectively analyzed patient data based on available Oncotype DX test results (RS) irrespective of nodal status at a single center. We collected recommendations from six oncologists, first without RS (pre-RS) and then with RS results (post-RS). We investigated changes in recommendations, agreement between oncologist decisions, and the effect of different National Comprehensive Cancer Network (NCCN) recommendation categories (for, against, and considering chemotherapy). Results: Data from 201 patients were included in the analysis. Recommendation of chemotherapy decreased by an average of 39.5%. Agreement improved substantially with RS, with a kappa value pre-RS of 0.37 (fair agreement) and post-RS of 0.75 (substantial agreement). Discordance remained substantial in cases where the NCCN recommendations considered chemotherapy only (32%). Pre-RS consensus against chemotherapy predicted low RS results (50 out of 51 patients). Post-RS consensus was highest in the NCCN chemotherapy recommendation group. Conclusions: The Oncotype DX test substantially improves decision accuracy in recommending adjuvant chemotherapy. It may be further improved with a consensus decision. In the case of pre-RS consensus against chemotherapy, the test can be spared.

## 1. Introduction

Breast cancer has the highest cancer incidence in women worldwide [1]. The treatment of the disease is a multidisciplinary task, which is especially true for early breast cancer. HR-positive (HR+) and HER2-negative breast cancers account for the vast majority of all breast cancers. In the treatment of HR+ HER2-negative early breast cancer, chemotherapy is an integral part of the complex treatment, but its benefit seems to be evident only for higher-risk tumors. Chemotherapy reduces the risk of recurrence, depending mainly on the characteristics of the tumor. In high-risk diseases, it significantly reduces the chance of recurrence and improves survival. In low-risk diseases, however, the added value is negligible, but in all cases, it is a significant burden for the patient (side effects, loss of employment, looser relationships with family and friends, and psychological stress) and may also have long-term consequences. The decision to give or not give adjuvant chemotherapy is the result of a complex process. It is influenced by several factors, such as the histopathological features and the stage of the tumor. In addition, the recommendation should depend on the Eastern Cooperative Oncology Group (ECOG) performance status scale, comorbidities, and the treatment preference of the patient [2]. Nevertheless, the final recommendation of adjuvant chemotherapy is also based on patient acceptance.

There is no “gold standard” method for choosing adjuvant chemotherapy. We cannot select the patient who will be cured by adjuvant chemotherapy. We can only determine which group of patients will benefit significantly from a particular treatment and which group of patients will not. The development and validation of multigene assays help identify patients who do not need adjuvant chemotherapy. Of the genomic tests, the 21-gene recurrence score (RS) assay (Oncotype DX) is one of the most widely used tests. The recurrence score is calculated from the expression of 16 cancer-related genes and 5 reference genes. Its value ranges from 0 to 100, and the higher the value, the higher the risk of recurrence. It predicts the likelihood of benefit from chemotherapy and the 10-year risk of distant recurrence, based on the results of several large clinical trials. The use of the Oncotype DX test reduces the rate of recommendation for adjuvant chemotherapy, reduces the financial burden and morbidity of complex therapy, and may improve survival at the same time [3]. Reducing the recommendation for chemotherapy and, more importantly, delivering chemotherapy more accurately are essential in adjuvant treatment. In a recent analysis involving 16 breast oncologists, Oncotype DX significantly increased the confidence of physicians in recommending adjuvant chemotherapy and the consensus of oncologists on the treatment to be given. [4]. However, even after the test has been performed, there remains some degree of uncertainty in the recommendation, especially in cases where professional recommendations are not clear. Premenopausal lymph node-negative patients with an RS of 16–25 or premenopausal lymph node-positive patients with an RS ≤ 25 may benefit from chemotherapy. This benefit may be a consequence of chemotherapy or chemotherapy-induced ovarian function suppression (OFS). The OFSET (NRG-BR009) phase 3 study addresses this question [5]. The trial compares the combination of adjuvant chemotherapy, OFS, and endocrine treatment (ET) to OFS plus ET.

The purpose of this real-world study was to assess how the results of the Oncotype DX test influence oncologist recommendations for adjuvant chemotherapy and the rate of actual adjuvant chemotherapy. It was particularly important to have data on the impact of the test on recommendations even after the results of the TAILORx and RxPONDER studies were published [6,7]. The TAILORx trial in patients with lymph node-negative cancer and the RxPONDER trial for patients with lymph node-positive cancer showed that, in general, adjuvant chemotherapy is not associated with a benefit in terms of invasive disease-free survival (IDFS) or overall survival (OS) if the RS is less than 26. Patients aged 50 years or younger may benefit from chemotherapy if there is no lymph node metastasis and an RS of 16–25, and if lymph node metastasis is present regardless of RS.

Our aim was to analyze oncologist recommendations before and after performing the Oncotype DX test. We included independent oncologists who were not involved in the care of the same patients to find trends in decision-making and associations with baseline characteristics. We also involved treating physicians in the validation of the model. We also aimed to identify situations where the test was most helpful for decision-making and those where it was least helpful.

## 2. Methods

Study Design. We collected relevant pathology results from all consecutive patients with hormone receptor-positive (HR+) (estrogen and/or progesterone receptor positive) and HER2 receptor-negative invasive tumors who underwent surgery at the National Institute of Oncology (NCI), Hungary in 2019, 2020 or 2022, and who had Oncotype DX results (recurrence score, RS). All patients had Oncotype DX, complete pathology results were available, and all patients were followed up at the NCI. Exclusion criteria were if the adjuvant treatment was given at another hospital, if the patient had breast cancer previously, or had other non-breast cancers in the past five years, if the patient had metastatic disease, if the patient had other serious illness or condition that excluded chemotherapy, and if more than 12 weeks had elapsed since surgery. We included five breast cancer specialists from one secondary and two tertiary oncology centers. All five specialists had more than five years of experience in breast cancer treatment, participated in multidisciplinary teams, and were consulted on more than 200 cases per year. Experts made recommendations first without knowledge of the RS (pre-RS), and then with knowledge of the RS (post-RS). We also asked the treating oncologists to retrospectively provide pre-RS recommendations with blinded data (post-RS recommendations of the treating physicians were collected from patient charts). For the retrospective evaluation, all patient data were anonymized for both experts and treating physicians. Patient factors (basic characteristics) were the following: age, grade, tumor stage (pathological primary tumor staging; pT in category and mm), lymph node status (pathological nodal status; pN category, the number of lymph nodes examined and the number of positivity), estrogen and progesterone receptor expression (in percentage), Ki67 in percentage, mitotic activity index (number of mitotic figures per 10 high-power fields), lymphocyte infiltration (TILs in percentage), vascular invasion (yes or no), and perinodal extension in lymph node-positive cases (yes or no). Patients with positive surgical margins and those who received adjuvant treatment at other hospitals were excluded from the analysis.

Statistical Analysis. We defined the agreement between oncologists as concordance when at least four out of five oncologists were in agreement. Complete concordance meant that all experts had the same opinion. In the rest of the cases, the common opinion was considered ambiguous. Agreement between oncologist decisions was measured using Fleiss’ kappa coefficient [8] (a variant of Cohen’s kappa [9] which can also be used to compare multiple experts). The agreement was rated as follows: slight agreement (κ = 0.00–0.20), fair agreement (κ = 0.21–0.40), moderate agreement (κ = 0.41–0.60), substantial agreement (κ = 0.61–0.80), and almost perfect agreement (κ = 0.81–1.00). We also analyzed the agreement between invited experts and treating physicians.

The chemotherapy recommendation is based on the results of the Oncotype DX RS according to the National Comprehensive Cancer Network Clinical Practice Guidelines in Oncology (NCCN Guidelines^®^) [10]. The Hungarian guideline is in line with the NCCN Guidelines [11]. The NCCN Guidelines for adjuvant chemotherapy recommendation distinguishes three categories: chemotherapy recommended (NCCN CT: RS ≥ 26), chemotherapy not recommended (NCCN no-CT: RS ≤ 25 in postmenopausal lymph node-positive or -negative cases, RS ≤ 15 in premenopausal lymph node-negative cases), and chemotherapy should be considered (NCCN cons: RS 16–25 premenopausal lymph node-negative cases, RS ≤ 25 in premenopausal lymph node-positive cases). Oncologists used these categories to formulate their opinions.

The associations between basic characteristics and expert recommendations were analyzed with logistic regression. We used the R statistical program version 4.3.3 [12]. The R package (version 0.84.1) was used to calculate interrater reliability data such as Kappa values.

The research was approved by the Central Ethics Committee (Medical Research Council, 21679-2/2016/EKU) and by the Regional Ethics Committee (14 March 2024).

## 3. Results

During the period studied, 204 patients had Oncotype DX results. Three patients were treated at other hospitals or had metastatic disease and were excluded from the analysis. The basic characteristics of patients are shown in Table 1. The mean age of patients was 52.9 years; most tumors were grade 2 (60%), and half of them were lymph node-positive. Low estrogen positivity was present in 2 cases, very high Ki67 (≥50%) in 5, and very high MAI (≥30) in 19. The mean RS value was 19.85 (80% ≤ 25 and 20% ≥ 26). Table 2 presents expert recommendations before and after knowledge of the RS results. There was a large variation in the reduction in recommending chemotherapy between experts (17.6–54.1%). The change from chemotherapy to no-chemotherapy recommendations (10–40.8%) and from no-chemotherapy to chemotherapy recommendations (1.5–16.4%) also varied substantially.

The level of consensus in the recommendations was substantially higher after the RS results (Table 3). The kappa value was 0.37 (fair agreement) before and 0.75 (substantial agreement) after the RS value was known to experts. We examined whether the opinions of the treating physicians differed more from those of the invited experts than from one expert to another. We found that this difference was very similar. The mean kappa value was 0.38 (range 0.31–0.45) when the treating physician was compared to each invited expert before RS, and the same value was 0.70 after RS (range 0.61–0.77). When we separately compared all invited experts to the others, the average kappa value was 0.34 before RS (range 0.19–0.49) and 0.75 after RS (range 0.64–0.86). The opinion of the treating physician was close to the average opinion of the invited experts (Figure 1 and Figure 2).

The relationship between NCCN categories and the consensus of expert recommendations was strongest when the NCCN clearly recommended chemotherapy (NCCN CT). The consensus was almost complete between experts, with only one expert voting against chemotherapy (Table 4). The divergence of opinions was the greatest when the NCCN recommended consideration (NCCN cons) of chemotherapy. The complete concordance rate was 98% in the NCCN CT, 89% in the NCCN no-CT, and 36% in the NCCN cons categories. The discordance remained substantial in the NCCN cons cases (32%). Restricted to the NCCN cons cases, the kappa value was 0.25 before knowing RS values, and 0.38 after knowing RS values, so despite the direct recommendation, the consensus slightly increased (both values were considered fair agreement).

After RS values were known to experts, 61 patients were recommended chemotherapy by the treating physicians and 61–90 by the invited experts. Chemotherapy was indicated based on expert consensus in 61 cases and by complete consensus in 50 cases. Of those initially recommended by the treating physician, a total of 45 started chemotherapy, 15 patients refused it, and one was found unsuitable for chemotherapy by the treating physician. The expert decision for these 16 patients was complete concordance against chemotherapy in 1, ambiguous in 1, concordance for chemotherapy in 3, and complete concordance in 11 cases.

We analyzed the pre-RS consensus association with the RS results and final decisions (Table 5). Expert opinions were complete consensus (all experts agreed) in 51 cases against chemotherapy. In all but one of these cases, the RS results were 25 or less and only one patient started chemotherapy. In detail, some of these patients had high-risk features (grade 3: 4, T3: 4, N+: 33, ER or PR ≤ 10%: 8, vascular invasion: 22 patients), and 13 of the 51 patients were under 50 years of age. Of these, eight were classified as “consider chemotherapy” according to the NCCN Guidelines, four with node-positive (RS: 6, 9, 17, and 20, respectively) and nine with node-negative disease (RS: 16–21: 3, RS: 22–25: 1).

Before RS results were known, expert recommendations differed substantially. The investigation of the association between recommendations of individual experts and basic characteristics indicated that each expert took some, but not all characteristics into account to a different extent (Table 6.). In the intermediate-risk group (NCCN cons group), no similar correlation was found after knowledge of the RS, with recommendations of only one expert (O_2_) associated with N status (*p* = 0.0014), T status, progesterone receptor and MAI (*p* = 0.05–>0.01).

## 4. Discussion

We examined the decision-making process for chemotherapy indication, and how it was influenced by Oncotype DX test results. In Hungary, the Oncotype DX test is available with specific permission from the health insurer. In the intermediate-risk population, the test is supported for all patients without restriction. For this purpose, we analyzed the recommendations of treating physicians and five invited senior oncologists before (pre-RS) and after (post-RS) Oncotype DX test results. In this patient population, the agreement between expert opinions for or against chemotherapy was initially fair and reached a higher level (substantial agreement) using the test (Fleiss’ kappa of 0.37 and 0.75, respectively). The overall rate of chemotherapy recommendations decreased by an average of 39.5% (23.7% of all patients). This means that an average of 56.4 (28.1%) of pre-RS planned chemotherapies were avoidable, and 14.2 (7%) additional chemotherapy recommendations emerged based on individual expert opinion. These changes, as well as the rates of initial chemotherapy recommendations (range 36.8–74.6%, average 60%), varied substantially between the experts, too. For example, chemotherapy recommendations decreased by 6.5–39.3% among individual oncologists. On average, the test led to a 35% change in recommendations. Oncologists 2 and 4 would have been characterized by overtreatment, and oncologists 3 and 5 would have been characterized by undertreatment without the Oncotype DX test. The overall discordance between opinions decreased from 28.9% to 11.4%. However, even after the Oncotpye DX test was performed, in 48 cases (23.9%) at least one oncologist made a recommendation different from those of the other oncologists, which could lead to over- or undertreatment in patients with the recommendation of a single oncologist. In terms of discordance, very similar results were published in a recent article by Licata et al. [4] where the inter-observer agreement was weaker before RS (K = 0.47) than after RS (K = 0.85), and the rate of discordance decreased from 27% to 7%. The chemotherapy recommendation also varied considerably before RS (range 26.7–76.7%) with an average rate of 51%, underlining the uncertainty in this intermediate-risk patient group. Both investigations showed that intra- and inter-observer differences were considerably decreased after the RS result became available. However, there were substantial differences between the investigations. Only 30 patients with lymph node-negative disease and 16 experts participated in the study by Licata. The experts were also asked to give recommendations at three levels of confidence: absolutely certain, fairly certain, or uncertain.

Licata found that uncertainty decreased in most cases after RSs, although it increased for 10% of the experts, showing that even genetic signatures could not always confirm expert confidence. In our investigation, involving 201 patients’ data (lymph node-positive and -negative cases) and the decisions of six oncologists for each patient case, we did not focus on expert confidence as a subjective element, but instead, we analyzed the importance of considering the opinions of more than one oncologist and compared it to the opinions of treating physicians. In cases where there was consensus or complete consensus of invited experts on chemotherapy, opinions substantially changed after RSs and less than half of the patients started chemotherapy. However, in cases of consensus or complete consensus against chemotherapy before RS, the test led to only a minor change in the recommendation and only 1 of 51 patients started chemotherapy. It suggests that not only genomic signatures but the consensus from more than one oncologist may make the decision more accurate, especially in the lower-risk population. This should be confirmed in other patient groups, but it could be a way of reducing costs in low-income countries. This is similar to initiatives using nomograms to identify patient groups who may avoid Oncotype testing in low-income countries [13,14,15,16,17]. The hypothesis that there are patient groups where the test could be spared is supported by results that show that RSs in grade I tumors are low in almost all cases. In a retrospective trial in Canada, none of the 236 patients with grade I tumors had a high-risk RS; however, in our dataset, 2 out of 10 patients with grade I tumors had RS results ≥26 [18]. In a retrospective analysis, the authors found that concordant low or high clinical features may predict RS results [19]. In contrast, there are conflicting results, too, showing no or moderate association between classical clinicopathological features and RSs [20,21,22]. Therefore, in spite of some promising results [23], the surrogate scores and nomograms have not become part of routine patient care.

Alkushi et al. published a similar analysis in 2021 [24], including data from 145 patients and pre-RS and post-RS recommendations from 16 oncologists. In this earlier trial, the risk group was defined differently (the so-called pre-TAILORx thresholds): low risk (RS ≤ 17), intermediate risk (RS = 18–30), and high risk (RS ≥ 31). The agreement between experts improved substantially, although not as much as in our study or in the trial by Licata (Fleiss kappa from 0.38 to 0.52). The smaller difference in agreement between experts is possibly due to the different RS thresholds used in the study by Alkushi (old limits of RSs) and in our investigation (new limits used in the TAILORx study), and the fact that the results of the RxPONDER study were published after the publication of Alkushi. Disagreement was highest in the intermediate-risk group (33.6%), and lower in the high-risk (31.3%) and low-risk groups (21.6%). Disagreement was also highest in the intermediate-risk group in our analysis (32%), although defined differently (according to NCCN Guidelines), and low in the high-risk (0%) and low-risk groups (4.8%). We observed substantial disagreement only in the intermediate-risk group. Before RS, we found different levels of correlation between basic characteristics and expert recommendations. Age, grade, lymph node status, and Ki67 had the largest impact on expert opinion, but similar correlations were not observed after RS, even in the intermediate-risk group. Interestingly, the opinion of the treating physician was weakly concordant with the opinion of the oncologist invited but better concordant with the average of the five opinions. This may be explained by the fact that several treating physicians gave opinions and, therefore, the opinions of the treating physicians also reflected different subjective elements.

The RS test and other genomic signatures result in significant changes in chemotherapy recommendations, which are well studied and generally of similar magnitude [25,26,27,28,29,30,31,32,33]. Previous studies with Oncotype DX used three risk categories based on RS: low (RS ≤ 17), intermediate (RS = 18–30), and high (RS ≥ 31). Current guidelines define the main threshold for RSs as 26. Below this value, chemotherapy is generally not recommended or may be considered for patients under the age of 50. Therefore, older study results may differ from recent ones. Furthermore, a chemotherapy recommendation is not equivalent to the actual initiation of chemotherapy. In our analysis, 16 patients (26%) did not receive chemotherapy despite strong recommendations. In two retrospective population-based studies, age and lymph node status, in addition to RS results, also influenced adjuvant chemotherapy [34,35,36]. In these populations with high RSs, only 64–66% of patients received chemotherapy. Moreover, after the age of 70, only about half of patients in the high-risk RS group received chemotherapy.

The main questions in adjuvant treatment planning are who will potentially benefit from a certain therapy and how the patient can benefit the most from a certain amount of the costs. The Oncotype DX test has been shown to be cost-effective in several analyses [37,38]. For a cost-effective study, it is important to have a good model. It is clear from our study and other studies that it is not enough to consider the opinion of a single oncologist and its change, because it is not representative of the differences between different perspectives, and it is not enough to measure the change in the rate of chemotherapy recommendation because it does not accurately reflect the actual number of patients receiving chemotherapy. Patients who refuse chemotherapy should be excluded from such analyses because their participation can substantially alter the estimate, and in addition, it is wasteful for them to have the test performed. In our example, excluding the 16 patients who refused chemotherapy (one-fourth of patients for whom chemotherapy was recommended), the indication for chemotherapy (consensus or complete consensus by oncologists invited) decreased from 82 (45%) pre-RS to 47 (25.8%) post-RS, which is a decrease of 57% instead of 39.5% in the whole cohort.

Genetic signatures, such as the Oncotype DX test, help clinicians choose more appropriate adjuvant treatment; they can be said to make clinicians “better or more senior experts”. We believe that a consensus decision may be even more appropriate. It can predict test results in lower-risk patients and can reduce discordance between the decisions of individual physicians even after the RS is known to them and help ensure that patients receive a uniformly appropriate recommendation. A remarkable uncertainty remains in some situations where guidelines do not provide a clear recommendation and advise consideration of chemotherapy based on the preference of the patient. Patient preference is always important, as in the end, patients bear the burden of side effects, but this should not influence professional recommendation. We believe that in these uncertain cases, a recommendation should be made that can be based on the consensus of a professional association (at the national level or according to the ASCO example [39]) that is accepted by the treating institutions. Of course, the level of evidence should also be discussed with patients before a final decision is made.

Our study has several limitations. The relatively modest sample size does not allow for a more detailed analysis in the dubious (NCCN cons) cohort. The retrospective nature of the investigation and the fact that only five experts from one country and three institutions took part in the investigation limits the generalizability of the results. The patients were selected from a single tertiary cancer center, and therefore, the opinion of the treating physician reflects the standard of care at a single institute. Due to the relatively short follow-up, no survival data were recorded; however, this was not the subject of the investigation.

## 5. Conclusions

The Oncotype DX test helps to make more adequate adjuvant treatment recommendations for patients with early breast cancer; it can make the oncologist a “more senior expert”. Furthermore, we believe that it may be more appropriate to involve multiple oncologists in adjuvant treatment planning, even if Oncotype DX or other multigene studies are planned. It may reduce the risk of incorrect recommendations. On the basis of our results, it seems that truly low-risk patients can be screened by consensus, which could be a cost-reducing factor and have a significant impact on the use of the test in lower-income countries. This result needs to be confirmed. Although Oncotype DX is indicated for 10–15% of patients with early breast cancer and only a quarter of them will have a questionable RS result without clear advice for or against chemotherapy, it is advisable that the relevant professional association or local working group make a recommendation for this 3–4% of patients with early breast cancer. The NRG-BR009 trial will hopefully elucidate the role of chemotherapy in this intermediate-risk population [5], and similarly, the introduction of new multigene tests may give more confidence in the decision-making process [40].

## Figures and Tables

**Figure 1 cancers-16-02928-f001:**
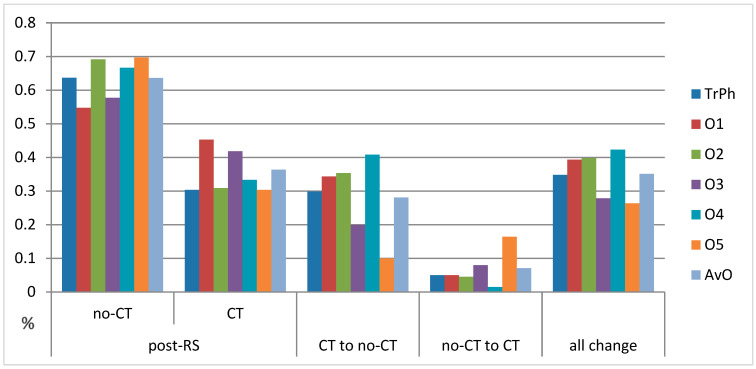
Proportion of recommendations and changes in recommendations. Recommendations of treating physicians were closest to the average of expert recommendations. all change—sum of no-CT to CT and CT to no-CT, AvO—average values of recommendations of the five invited oncologists, no-CT/CT—does not recommend/recommend chemotherapy, O1–O5—invited oncologists 1–5, post-RS—knowing the RS result, TrPh—treating physician.

**Figure 2 cancers-16-02928-f002:**
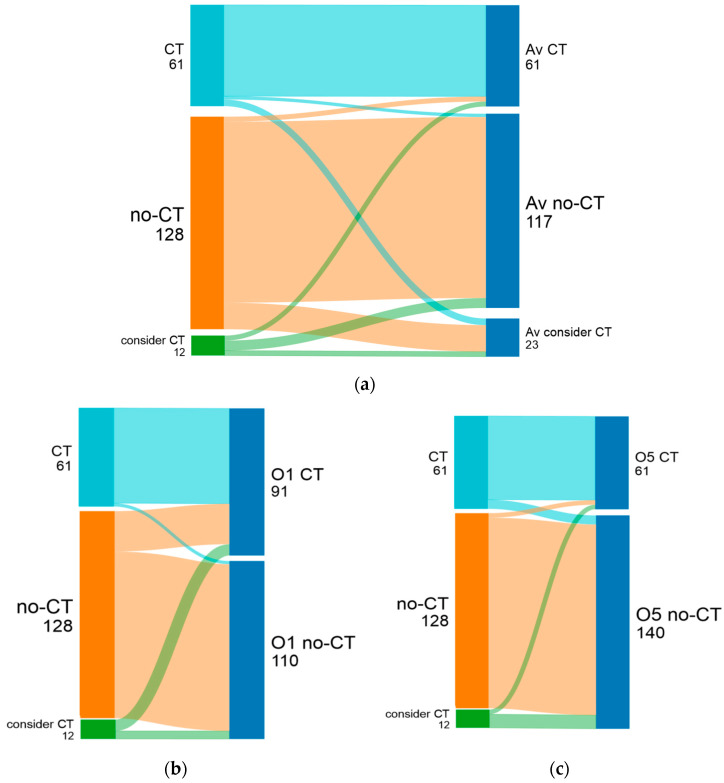
Post-RS chemotherapy recommendations by treating physicians (left) and (**a**) average of all invited oncologists (Av, right), (**b**) oncologist 1 (less concordance), and (**c**) oncologist 5 (best concordance). In Sankey plots, thickness of lines indicates number of patients.

**Table 1 cancers-16-02928-t001:** Basic characteristics.

Characteristic	Mean (min.-max. Value)	Categories (%)
age (year)	52.9 (25–75)	
<50 year		86 (43%)
≥50 year		115 (57%)
grade I		10 (5%)
grade II		122 (60.1%)
grade III		69 (34.3%)
pT (mm)	26 (2.8–85)	
1a		1 (0.5%)
1b		2 (1%)
1c		79 (39.3%)
2		106 (52.7%)
3		13 (6.5%)
pN		
0		102 (50.7%)
1		98 (48.8%)
2		1 (0.5%)
ER (%)	94 (1–100)	
PR (%)	59 (0–100)	missing: 1
Ki67 (%)	20 (<1–60)	
MAI	12.5 (1–65)	missing: 6
N^o^ positive lymph node	0.73 (0–5)	
N^o^ excised lymph nodes	4.16 (1–31)	
perinodal infiltration		
yes		62 (30.8%)
no		25 (12.4%)
missing		12 (6%)
TILs (%)	5.8 (0–70)	missing: 24 (11.9%)
vascular invasion		
yes		90 (44.8%)
no		102 (50.7%)
missing		10 (5%)
NPI	4.31 (2.5–5.7)	
RS	19.85 (0–66)	
0–10		33 (16.4%)
11–15		31 (15.4%)
16–20		48 (23.9%)
21–25		48 (23.9%)
26–30		14 (7%)
>30		27 (13.4%)
adjuvant chemotherapy		
yes		46 (22.9%)
no		155 (77%)

ER—estrogen receptors, grade—histologic grade, MAI—mitotic activity index, NPI—Nottingham Prognostic Index, pT—pathologic tumor status and size, pN—pathologic lymph node status, PR—progesterone receptor, RS—recurrence score, TILs—tumor-infiltrating lymphocytes.

**Table 2 cancers-16-02928-t002:** Change in adjuvant chemotherapy recommendations.

Oncologist	Pre-RS		Post-RS		CT Recommendation Decrease (Percentage of All Patients)	Change from CT to No-CT	Change from No-CT to CT	All Changes
	No-CT	CT	No-CT	CT				
treating physician	82 (41%)	119 (59%)	128 (63.7%)	61 (30.3%)	48.7% (28.9%) uncertain after RS:12 (6%)	60 (29.9%)	10 (5%)	70 (34.9%)
oncologist 1	51 (25.4%)	150 (74.6%)	110 (54.7%)	91 (45.3%)	35% (26.4%)	69 (34.3%)	10 (5%)	79 (39.3)
oncologist 2	77 (38.3%)	124 (61.7%)	139 (69.2%)	62 (62%)	50% (30.8%)	71 (35.3%)	9 (4.5%)	80 (39.8%)
oncologist 3	92 (45.8%)	109 (54.2%)	116 (57.7%)	84 (41.8%)	22.9% (12.4%)	40 (19.9%)	16 (8%)	56 (27.9%)
oncologist 4	55 (27.4%)	146 (72.6%)	134 (66.7%)	67 (33.3%)	54.1% (39.3%)	82 (40.8%)	3 (1.5%)	85 (42.3%)
oncologist 5	127 (63.2%)	74 (36.8%)	140 (69.7%)	61 (30.3%)	17.6% (6.5%)	33 (16.4%)	20 (10%)	53 (26.4%)
average of oncologists 1–5	80.4 (40%)	120.6 (60%)	127.8 (63.6%)	73 (36.3%)	39.5% (23.7%)	56.4 (28.1%)	14.2 (7%)	70.6 (35%)

CT—chemotherapy, no-CT—no chemotherapy, pre-RS—recommendation before RS result, post-RS—recommendation after RS result.

**Table 3 cancers-16-02928-t003:** Variation in the agreement of the five experts.

Agreement Level	Pre-RS		Post-RS	
complete concordance	74 (36.8%)	no-CT: 23 (11.4%)	153 (76.1%)	no-CT 103 (51.2%)
		CT: 51 (25.4%)		CT 50 (24.9%)
concordant	69 (34.3%)	no-CT: 28 (13.9%)	25 (12.4%)	no-CT 14 (7%)
		CT: 41 (20.4%)		CT 11 (5.4%)
discordance	58 (28.9%)		23 (11.4%)	

Concordant—at least four oncologists were of the same opinion, complete concordance—all five were of the same opinion, discordant—only three gave the same recommendation.

**Table 4 cancers-16-02928-t004:** Association of NCCN recommendation category and expert opinion knowing RS values (concord 0—all five experts recommended against chemotherapy, concord 5—all five experts recommended chemotherapy).

NCCN Category (Number of Patients)	Concord 0	Concord 1	Concord 2	Concord 3	Concord 4	Concord 5
no-CT (*n* = 104)	93	6	1	4	0	0
CT (*n* = 41)	0	0	0	0	1	40
consider CT (*n* = 56)	10	8	11	7	10	10

**Table 5 cancers-16-02928-t005:** Association of original expert consensus, RS result, and adjuvant chemotherapy.

Pre-RS Test Opinion for Chemotherapy by Experts	Number	RS Result	Final Decision of Experts	Chemotherapy Was Given
CCA	23	all RS ≤ 24	CCA: 23 CA: 1 ambiguous: 1	1
CA	28	RS ≤ 25: 27 RS ≥ 26: 1	CCA: 24 CA: 3 ambiguous: 1 CCF: 1	0
CF	41	RS ≤ 25: 34 RS ≥ 26: 7	CCA: 16 CA: 4 ambiguous: 7 CF: 3 CCF: 11	9
CCF	51	RS ≤ 25: 26 RS ≥ 26: 25	CCA: 9 CA: 2 ambiguous: 5 CF: 4 CCF: 30	26

CCA—complete concordance against (5/5 opinions) chemotherapy, CA—concordance against (4/5 opinions) chemotherapy, CF—concordance for (4/5 opinions) chemotherapy, CCF—complete concordance for (5/5 opinions) chemotherapy.

**Table 6 cancers-16-02928-t006:** Association between basic characteristics and expert recommendation before knowledge of RS (logistic regression results using listed characteristics as independent variables).

	Age	Grade (1–2 vs. 3)	T (1 vs. 2–3)	N (0 vs. 1)	ER (≥30% vs. <30%)	PR (≥30% vs. <30%)	Ki67 (≥20% vs. <20%)	MAI (≥20 vs. <20)	Vascular Invasion (Yes vs. No)
O1		**					***		
O2	***	*	***	***		*	***		
O3	***	***	*	**			***		
O4	***					*	***		**
O5	*	***		***	*	*	***	***	**

Level of significance (*p*): * 0.05–>0.01, ** 0.01–>0.001, and *** ≤0.001.

## Data Availability

A blinded dataset is available upon request. It is currently in Hungarian but can be translated upon request.

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
