# Peer review of "How to Tackle Discordance in Adjuvant Chemotherapy Recommendations by Using Oncotype DX Results, in Early-Stage Breast Cancer"

_cancers, 2024, doi:10.3390/cancers16172928_

Round 1
Reviewer 1 Report
Comments and Suggestions for Authors
This paper shows that the result of the Oncotype test changes the recommendation for adjuvant chemotherapy, but does not prove that this change is beneficial. It seems cost-effective for the payor and the real value appears positive from other publications, but not from this paper.
I recommend to emphasize this point.
Author Response
R1: This paper shows that the result of the Oncotype test changes the recommendation for adjuvant chemotherapy, but does not prove that this change is beneficial. It seems cost-effective for the payor and the real value appears positive from other publications, but not from this paper.
I recommend to emphasize this point.
We gratefully thank the reviewer for his comment. We emphasized the importance of the Oncotype DX test for appropriate chemotherapy recommendations and its benefits for patients. The Oncotype DX test not only reduced the number of chemotherapy recommendations but also changed the recommendation to chemotherapy in about 7% of patients. From our data, it appears that expert consensus may make recommendations more appropriate. Accordingly, we modified the Introduction and Discussion sections.
Reviewer 2 Report
Comments and Suggestions for Authors
The manuscript is well written and addresses an important issue in decision-making for patient benefit. The authors have conducted thorough research and have clearly presented their study results and methods. I enjoyed reading the manuscript, and it is excellent in its current form.
I have only one comment: How often is the DX test recommended when deciding on chemotherapy? Is it required or optional? Additionally, how often is it covered by insurance? These factors can significantly influence a physician's likelihood of recommending Oncotype DX for patients and how frequently they refer to it. Perhaps addressing these points can further improve the impact of the manuscript.
Author Response
I have only one comment: How often is the DX test recommended when deciding on chemotherapy? Is it required or optional? Additionally, how often is it covered by insurance? These factors can significantly influence a physician's likelihood of recommending Oncotype DX for patients and how frequently they refer to it. Perhaps addressing these points can further improve the impact of the manuscript.
We gratefully thank the reviewer for reviewing our manuscript. In Hungary, the Oncotype DX test is recommended for medium-risk patients. The examination is not mandatory, but it is professionally highly recommended and covered by health insurance. We have modified the Discussion section: “In Hungary, the Oncotype DX test is available with specific permission from the health insurer. In intermediate-risk population, the test is supported for all patients without restriction.”
Reviewer 3 Report
Comments and Suggestions for Authors
Manuscript: How to Tackle Discordance in Adjuvant Chemotherapy 2 Recommendations by Using Oncotype DX Results, in Early Stage Breast Cancer
In this study, the authors aimed to investigate the decision-making process for chemotherapy (CT) indications and how it is influenced by Oncotype DX test results in patients with hormone receptor-positive (HR+) (estrogen and/or progesterone receptor-positive) and HER2 receptor-negative breast cancer tumors. They analyzed the recommendations of treating physicians and five invited senior oncologists before (pre-RS) and after (post-RS) obtaining the Oncotype DX results. The manuscript has excellent technical quality, and the results and discussion are well-written. The clinical limitations of the study findings were well presented at the end of the discussion section.
I do have some comments and suggestions:
1. Abbreviations are usually defined at the first use in the abstract. Please expand the ‘NCCN’ abbreviation in line 28.
2. In the introduction, the authors provide a concise and clear background for their objectives. However, I strongly recommend including a couple of sentences explaining what the present manuscript adds to the existing literature regarding the Oncotype DX assay in guiding oncologists on chemotherapy recommendations.
3. For better comprehension, the authors should divide the Methodology section into two subsections: 'Study Design' and 'Statistical Analyses.' In the Statistical Analyses subsection, please provide the R version and the R packages used to perform the analysis.
4. Please include the inclusion and exclusion criteria for the study population in the Methodology section.
5. In the Results section, I recommend that the authors include the menopause status of all volunteers in Table 1.
6. In Figure 1, please add the y-axis legend and define the abbreviations for the color legend after the figure caption.
7. There are several mentions of the difference between the agreement in CT recommendations before and post-RS as 'statistically significant.' However, no additional test was performed to measure statistical significance. The term 'significant' should be followed by a p-value. Please determine the p-value to assess the significance or replace the term with 'remarkable,' 'substantial,' or other related terms.
Author Response
- Abbreviations are usually defined at the first use in the abstract. Please expand the ‘NCCN’ abbreviation in line 28.
We gratefully thank the reviewer for reviewing our manuscript. We inserted the expanded name of NCCN.
- In the introduction, the authors provide a concise and clear background for their objectives. However, I strongly recommend including a couple of sentences explaining what the present manuscript adds to the existing literature regarding the Oncotype DX assay in guiding oncologists on chemotherapy recommendations.
We gratefully thank the reviewer for his comment. Our investigation is the second, which was performed involving several oncologists and using new RS limit values, and it is the first which involved lymph node-positive patients, too. Furthermore, our data showed that – beyond each oncologist’s recommendations becoming more appropriate with the test - it is beneficial if more than one oncologist makes a recommendation even after doing the Oncotype DX test. We amended the discussion section and emphasized these points.
- For better comprehension, the authors should divide the Methodology section into two subsections: 'Study Design' and 'Statistical Analyses.' In the Statistical Analyses subsection, please provide the R version and the R packages used to perform the analysis.
We sincerely thank you for the comment. We have made these corrections in the manuscript.
- Please include the inclusion and exclusion criteria for the study population in the Methodology section.
We sincerely thank you for the comment. We inserted inclusion and exclusion criteria in the manuscript.
- In the Results section, I recommend that the authors include the menopause status of all volunteers in Table 1.
Thank you for your comment and we agree with it. Unfortunately, due to the retrospective nature of the study, we did not find data on menopausal status for many patients. However, in the TAILORx trial, there was no interaction between menopausal status and chemotherapy benefit, and in the RxPONDER trial, no chemotherapy benefit was observed in premenopausal patients aged 50 years and older. In both trials, chemotherapy was associated with some chemotherapy benefit in patients younger than 50 years (in cases if RS was 16-25 for lymph node-negative patients and if RS was below 26 for lymph node-positive patients). Consequently, in our opinion patient age is taken into account to a greater extent and menopausal status is less important in decision-making. We therefore decided not to present incomplete data on menopausal status.
- In Figure 1, please add the y-axis legend and define the abbreviations for the color legend after the figure caption.
Thank you for your comment. We have corrected Figure 1.
- There are several mentions of the difference between the agreement in CT recommendations before and post-RS as 'statistically significant.' However, no additional test was performed to measure statistical significance. The term 'significant' should be followed by a p-value. Please determine the p-value to assess the significance or replace the term with 'remarkable,' 'substantial,' or other related terms.
Thank you for the comment. When we used the word “significant” we didn’t mean “statistical significance”, however, we understand that it may lead to misunderstanding. Therefore, we thank your remark and we have changed “significant” to other words.
Reviewer 4 Report
Comments and Suggestions for Authors
How to Tackle Discordance in Adjuvant Chemotherapy Recommendations by Using Oncotype DX Results, in Early Stage Breast Cancer
This study focus on the investigation of the changes of recommendation from five oncologists and one treating physician to evaluate patients with or without the chemotherapy before and after reviewing the RS result. However, there are lots of gramma issues in the entire manuscript which making it difficult to understand the purpose, result and the conclusion. Some statements are also confused or not convincing enough due to the lack of providing enough background information, clear explanation of experimental design and purpose. Since there are publications showing the advantages of Oncotype DX results with increase concordance in adjuvant chemotherapy recommendations for early-stage breast cancer, what is importance of this study to benefit the work for the early stage breast cancer? Highly recommend the authors to rewrite the overall manuscript (checking gramma) and highlight the importance/purpose of this study. The population of the patients is relative large in this studies, however the number of the specialists is small (5 specialists) and they are from same country, so the opinions from these specialists could be representative only in this certain area. However, since the medical level varies from country to country, will the results be more convincing by including expert representatives from different countries?
Comments:
Line 24, for remaining questions, please specify them.
Line 25, for Oncotype DX test results (RS), please include what RS stands for and the score range to predict the likelihood of breast cancer returning.
Line 44, for the introduction part, will it be better to include the advantages or disadvantages of the adjuvant chemotherapy? Such as what happens if patients are treated with adjuvant chemotherapy but they are actually not necessary need to? The cost and the patient cognition of adjuvant chemotherapy?
Line 48-50, please rewrite the whole sentence.
Line 53, Do you mean the ECOG Performance Status Scale? What ECGO stands for?
Line 61, what does the Oncotype DX test do? How it work on analyzing samples?
Line 64-66, please rewrite this whole sentence, missing subject.
Line 67-70, please break down this long sentence to make it clear. What are TAILORx and RxPONDER studies? Background information about these studies with the adjuvant chemotherapy on patients is lacking here, making it confusing.
Line 70-72, line 77-76, line 84-85, line 116, gramma issue, please rewrite those whole sentence. There are a lot of gramma issues detected in the whole presentation, making it difficult to understand the goal, result and the conclusion of this study.
Line 86, line 87, please provide the full name of pT and pN.
Line 112, please specify what statistic data was collected for the log-rank test and R statistical program.
Line 133-143, for the discussion in this part, the level of consensus in recommendations between oncologists and treating physician were compared. Questions are: The oncologists and treating physician might have different areas of expertise and roles in a patient's care, what is the purpose on this comparison? In addition, the result is based on only one treating physician, can this result represent the average? Comparing the result from oncologists and treating physician, which result is more important for this study?
Line 130, in table 2, in the post-RS result from treating physician, why total patient number for no-CT and CT is less than 201? Also the percentage changes from CT to no-CT showed the huge difference among five oncologists, is it because they trust the RS in different levels? What is the effective ways to avoid the under –and over treatment of some patients? With the RS included in the decision making among oncologists, does it help to improve the confidence for patients to accept the treatment?
Line 150-154, please provide more details in the figure description to make it clear, such as what does that mean when the green lines are in different thickness. Also, what is the purpose to display the result of the oncologist 1 and oncologist 5 to treating physician? Wouldn’t it better to compare the Pre-RS and Post-RS?
Line 247-249, is it possible that the smaller difference in the agreement between experts is due to the smaller number of experts from this study compared to the study from Alkushi et al?
Line 254-257, regarding to those statements, some questions are: why it is important to compared one treating physician to the average of the opinion from 5 oncologists? There will be high possibility that the selected treating physician just provided a result with high correlation with the average of opinions from 5 oncologists. If other treating physician was chosen, will the correlation be different? Please specify the purpose and explain why the result from one treating physician is enough to conclude the statement.
Line 315-319, if the population size increase, there will be more patients having chances to get the questionable RS result, then there will more extra work be needed to professionals to make recommendations. Will this be also concerns for using the Oncotype DX Results?

Please see the attached.
Author Response
This study focus on the investigation of the changes of recommendation from five oncologists and one treating physician to evaluate patients with or without the chemotherapy before and after reviewing the RS result. However, there are lots of gramma issues in the entire manuscript which making it difficult to understand the purpose, result and the conclusion. Some statements are also confused or not convincing enough due to the lack of providing enough background information, clear explanation of experimental design and purpose. Since there are publications showing the advantages of Oncotype DX results with increase concordance in adjuvant chemotherapy recommendations for early-stage breast cancer, what is importance of this study to benefit the work for the early stage breast cancer? Highly recommend the authors to rewrite the overall manuscript (checking gramma) and highlight the importance/purpose of this study. The population of the patients is relative large in this studies, however the number of the specialists is small (5 specialists) and they are from same country, so the opinions from these specialists could be representative only in this certain area. However, since the medical level varies from country to country, will the results be more convincing by including expert representatives from different countries?
We sincerely appreciate the whole review and all the comments. We have amended the introduction to describe the purpose of the research in more detail. We believe this study is important because it shows that consensus opinion is more accurate even after Oncotype DX testing and therefore it is reasonable to consider the combined opinion of several physicians when making a decision. Furthermore, consensus against adjuvant chemotherapy before the Oncotype DX trial may be an option to reduce costs in low-income countries.
We have reviewed the manuscript again and again and made corrections to ensure that the statements are clear and rationally based. The manuscript was also proofread by a professional and several grammatical errors have been corrected.
We agree that the results would be more generalizable if more oncologists from more countries were included. We plan to continue the research and include more oncologists from other countries. However, we believe that the method used was acceptable to demonstrate the importance of consensus and could form the basis for further studies in this area.
Comments:
Line 24, for remaining questions, please specify them.
Thanks for the comment. There is no more space to explain the topic, so we deleted it from the Abstract and described it in the Introduction section.
Line 25, for Oncotype DX test results (RS), please include what RS stands for and the score range to predict the likelihood of breast cancer returning.
Thanks for the comment. We inserted it in the introduction section.
Line 44, for the introduction part, will it be better to include the advantages or disadvantages of the adjuvant chemotherapy? Such as what happens if patients are treated with adjuvant chemotherapy but they are actually not necessary need to? The cost and the patient cognition of adjuvant chemotherapy?
Thank you for the comment. We inserted it in the introduction section.
Line 48-50, please rewrite the whole sentence.
Thank you for the comment, we rewrote the sentence.
Line 53, Do you mean the ECOG Performance Status Scale? What ECGO stands for?
Thank you for the comment, we corrected it.
Line 61, what does the Oncotype DX test do? How it work on analyzing samples?
Thank you for the comment; we inserted essential facts about the test in the introduction section.
Line 64-66, please rewrite this whole sentence, missing subject.
Thank you for the comment; we corrected the sentence.
Line 67-70, please break down this long sentence to make it clear. What are TAILORx and RxPONDER studies? Background information about these studies with the adjuvant chemotherapy on patients is lacking here, making it confusing.
Thank you for the comment; we have made the proposed modification and added the text.
Line 70-72, line 77-76, line 84-85, line 116, gramma issue, please rewrite those whole sentence. There are a lot of gramma issues detected in the whole presentation, making it difficult to understand the goal, result and the conclusion of this study.
Thank you for the comment. We requested proofreading in English and corrected the text accordingly.
Line 86, line 87, please provide the full name of pT and pN.
Thank you for the comment. We corrected the text.
Line 112, please specify what statistic data was collected for the log-rank test and R statistical program.
Thank you for your comments. These characteristics are listed as patient factors in the methodology section. We have inserted the term "basic characteristics" there to make it clear.
Line 133-143, for the discussion in this part, the level of consensus in recommendations between oncologists and treating physician were compared. Questions are: The oncologists and treating physician might have different areas of expertise and roles in a patient's care, what is the purpose on this comparison? In addition, the result is based on only one treating physician, can this result represent the average? Comparing the result from oncologists and treating physician, which result is more important for this study?
Thank you for the questions. The investigated 201 patients had several treating physicians, 10 altogether. The treating physicians may have different level of expertise, however, all treating physicians in our institution has remarkable expertise in treating breast cancer patients. More than eight hundred primary breast cancers are operated yearly. There are differences between the situations when a physician responsible for patient care gives a recommendation or when an oncologist gives a recommendation retrospectively. The latter is common in retrospective analyses evaluating the effect of the Oncotype test. We think that involving the treating physicians in the analysis makes the results more generalizable.
Line 130, in table 2, in the post-RS result from treating physician, why total patient number for no-CT and CT is less than 201? “uncertain after RS:12 (6%)” Also the percentage changes from CT to no-CT showed the huge difference among five oncologists, is it because they trust the RS in different levels? What is the effective ways to avoid the under –and over treatment of some patients? With the RS included in the decision making among oncologists, does it help to improve the confidence for patients to accept the treatment?
Thank you for your comments and questions. The recommendations of the attending physicians after RS were collected from the patient's documentation. In the case of the treating physician's recommendations, uncertainty was documented in the patient charts in 12 cases.
The change in recommendations was different for the 5 oncologists. This is probably because their recommendations differed significantly before RS, and some of them recommended much less chemotherapy before RS. Among them, the decrease in chemotherapy recommendations was less pronounced.
Performing the Oncotype test considerably reduces the amount of undertreatment or overtreatment. In our study, consensus after RS may further reduce this measure.
If patients are familiar with the Oncotype test, the test certainly improves their confidence in accepting chemotherapy, but in our experience, patients turned to doctors as experts with great confidence even before the Oncotype test was available.
Line 150-154, please provide more details in the figure description to make it clear, such as what does that mean when the green lines are in different thickness. Also, what is the purpose to display the result of the oncologist 1 and oncologist 5 to treating physician? Wouldn’t it better to compare the Pre-RS and Post-RS?
Thank you for your comments. We have changed the description of the figure.
With this figure, we would like to present the different changes in the oncologists' recommendations, as well as to present examples for comparison where the opinions were relatively close or far from each other. We believe that this may also emphasize the importance of consensus-based recommendations.
Line 247-249, is it possible that the smaller difference in the agreement between experts is due to the smaller number of experts from this study compared to the study from Alkushi et al?
Thank you for your questions. We think that the smaller difference in the agreement between experts is possibly due to the different RS value thresholds used in Alkushi’s (old limits of RS) and in our investigation (new limits used in TAILORx study), and the fact that RxPONDER study results were published after Alkushi’s publication.
Line 254-257, regarding to those statements, some questions are: why it is important to compared one treating physician to the average of the opinion from 5 oncologists? There will be high possibility that the selected treating physician just provided a result with high correlation with the average of opinions from 5 oncologists. If other treating physician was chosen, will the correlation be different? Please specify the purpose and explain why the result from one treating physician is enough to conclude the statement.
Thank you for your question. When we mentioned treating doctors, we were referring to the doctor of the given patient. 10 different treating physicians participated in our analysis. As we answered a previous question, there are differences between the situations when a physician responsible for patient care gives a recommendation or when an oncologist gives a recommendation retrospectively. The latter is common in retrospective analyses evaluating the effect of the Oncotype test. We think that involving the treating physicians in the analysis makes the results more generalizable.
Line 315-319, if the population size increase, there will be more patients having chances to get the questionable RS result, then there will more extra work be needed to professionals to make recommendations. Will this be also concerns for using the Oncotype DX Results?
Thank you for the question. We think that consensus recommendation can be beneficial irrespective of RS result. However, in the case of high RS the consensus was very high, and in these cases not important to involve more oncologists in the decision-making. The disagreement was also relatively low in the RS low-risk group (4.8%), but high in the RS intermediate-risk group (32%). For them, consensus decision-making seems to be reasonable and beneficial. We don’t think that this may have an impact on Oncotype use.
Reviewer 5 Report
Comments and Suggestions for Authors
The authors of this paper put a significant effort into highlighting the challenges and controversies of choosing adjuvant treatment in ER+ early breast cancer patients. There are no flaws in the methodology, and the limitations of the research are clearly defined by the authors.
Two minor points need correction. First, the percentage of Oncologist 2 in Table 2 post-RS needs to be corrected. Second, you indicate the NRG-BR009 trial at the conclusion without any prior description of it. Given its great importance and anticipated results, a comment about it is needed earlier in the text.
Comments on the Quality of English LanguageIn some parts (mainly in the introduction) of the manuscript, English can be improved, mainly with syntax editing.
Author Response
Two minor points need correction. First, the percentage of Oncologist 2 in Table 2 post-RS needs to be corrected. Second, you indicate the NRG-BR009 trial at the conclusion without any prior description of it. Given its great importance and anticipated results, a comment about it is needed earlier in the text.
We gratefully thank you your review and your comments. In Table 2 we double checked data of Oncolgist 2 and we didn’t find incorrect data.
We added NRG-BR009 trial details in the Introduction section.
In some parts (mainly in the introduction) of the manuscript, English can be improved, mainly with syntax editing.
Thank you for the comment. We requested proofreading in English and corrected the text accordingly.
Reviewer 6 Report
Comments and Suggestions for Authors
In this study entitled ‘How to Tackle Discordance in Adjuvant Chemotherapy Recommendations by Using Oncotype DX Results, in Early 3 Stage Breast Cancer’, the use of the OncoType DX Recurrence Score Assay test was evaluated in the context of adjuvant chemotherapy decisions. Data from 201 patients were used, involving six oncologists together with the treating oncologist. This work is worth adding to the literature with minor corrections.
Introduction:
The TAILORx and RxPONDER trials can be explained in one sentence to give readers an overview.
A brief information about the genes in the assay should be given.
Discussion:
The data obtained are valuable. The discussion needs to be revised to make it more appealing.
In my opinion, the data should be better correlated with clinical characteristics, here are some: HR-positive patients were selected. Hormone positivity could be separated as ER-positive and PR-positive in the discussion. TIL and vascular invasion should be correlated with the results. Lymph node status should also be considered.
Author Response
Introduction:
The TAILORx and RxPONDER trials can be explained in one sentence to give readers an overview.
A brief information about the genes in the assay should be given.
We gratefully thank you for the review and your comments. We added basic information about TAILORx and RxPONDER trials, and also about genes of the Oncotype test in the Introduction section.
Discussion:
The data obtained are valuable. The discussion needs to be revised to make it more appealing.
In my opinion, the data should be better correlated with clinical characteristics, here are some: HR-positive patients were selected. Hormone positivity could be separated as ER-positive and PR-positive in the discussion. TIL and vascular invasion should be correlated with the results. Lymph node status should also be considered.
Thank you for your valuable remarks. We focus on the associations of basic characteristics and oncologist recommendations, since the investigation of correlation between basic characteristics and RS results were not among the purposes of the study and it is well studied that the correlation is week. As all tumors were ER+ and only 17 were PR negative, we used the 30% threshold for hormone receptor levels for the analyses. These results were amended in the Discussion section: “Before RS, we found different levels of correlation between basic characteristics and expert recommendations. Age, grade, lymph node status, and Ki67 had the largest impact on expert opinion, but similar correlations were not observed after RS, even in the intermediate-risk group.”
Round 2
Reviewer 4 Report
Comments and Suggestions for Authors
Thank you for answering all my questions. With more details included in the introduction, more explanations about the experimental design, it makes the purpose of this study more clear. The statement becomes clearer and more rational. I don't have any further questions.